

# CircVAMP3 promotes acute lung injury progression by regulating the miR-580-3p/STING/IFN-β axis in macrophages

Zheng Zhou[*], Shuyang Chen[*], Yajun Wang, Shujing Chen and Jinjun Jiang

Department of Pulmonary and Critical Care Medicine, Zhongshan Hospital, Fudan University, Shanghai, China

[*] These authors contributed equally to this work.

## ABSTRACT

Circular RNA (circRNA) is a prominent group of non-coding RNA (ncRNA) that regulates gene expression at the post-transcriptional level in various diseases including acute respiratory distress syndrome (ARDS). In our study, circVAMP3 was a novel circRNA significantly increased in macrophages in an acute lung injury (ALI) model. Through knockdown and overexpression of circVAMP3, we found circVAMP3 promoted the expression of IFN-β in macrophages, which was vital in the pathophysiological process of ARDS. Moreover, we showed that circVAMP3 targeted Stimulator of Interferon Genes (STING) through miR-580-3p by bioinformatics analysis and intervene of miR-580-3p. CircVAMP3 acts as a pro-inflammation ncRNA by regulating the miR-580-3p/STING/IFN-β axis and could be a potential biomarker and therapeutic target for ARDS.

Corresponding authors
Shujing Chen,
chen.shujing1@zs-hospital.sh.cn
Jinjun Jiang, jinjundoc@163.com

## INTRODUCTION

Circular RNAs (circRNAs) represent a prominent group of non-coding RNAs (ncRNAs) characterized by their circular structure, which confers enhanced stability compared to traditional linear RNA molecules. This unique conformation enables circRNAs to resist exonuclease-mediated degradation, thereby maintaining prolonged intracellular activity (*Chen & Yang, 2015*; *Jeck et al., 2013*). Through alternative splicing mechanisms, substantial quantities of circRNAs are generated, each exhibiting distinct sequence compositions and spatiotemporal expression patterns (*Maass et al., 2017*; *Xia et al., 2017*).

Functionally, circRNAs serve as competitive endogenous RNAs (ceRNAs) through their miRNA sponge activity. By sequestering microRNAs (miRNAs), these circular transcripts alleviate miRNA-mediated suppression of target mRNAs (*Hansen et al., 2013*), effectively modulating post-transcriptional gene regulation. The intricate circRNA-miRNA-mRNA regulatory network participates in diverse cellular processes, while extracellular circRNAs can exert paracrine/endocrine effects through intercellular communication, influencing both local and distal cellular activities (*Li et al., 2015*).

Acute lung injury (ALI) and acute respiratory distress syndrome (ARDS) are characterized by diffuse alveolar damage in the acute inflammatory phase, manifesting as epithelial cell death, disrupted endothelium, damaged mitochondria, impaired fluid and ion clearance, activation of resident immune cells and recruited circulating immune cells, which ultimately results in respiratory failure (*Matthay et al., 2019*). Pneumonia, non-pulmonary sepsis, aspiration, trauma and many other causes may lead to the development of ALI/ARDS which is associated with high mortality and long-term complications. Due to complex etiology and histology changes, the pathophysiological mechanism of ALI/ARDS remain incompletely understood.

Emerging evidence highlights the clinical potential of circRNAs as biomarkers for ARDS. The high expression level of circ_2979 in ARDS patients indicates its potential diagnostic and prognostic value (*Xu et al., 2023*). The overexpression level of circN4bp1 in immune cells is correlated with the prognosis of ARDS patients caused by sepsis (*Zhao et al., 2021*). Despite these advances, numerous circRNAs remain to be comprehensively characterized, warranting further investigation into their mechanistic roles and translational applications.

Hence, our study aims to identify clinically actionable circRNAs in ALI/ARDS for diagnostic and therapeutic development, and decode type 1 interferon regulatory mechanisms to discover targeted therapies.

## MATERIALS & METHODS

### Cell culture and treatment

THP-1 cells, a human monocytic cell line, were obtained from Fuheng (Guangzhou, China) and cultured in the Roswell Park Memorial Institute-1640 (Cat. No. 11875093 Gibco, Waltham, MA, USA) medium supplemented with 10% fetal bovine serum (Cat. No. F0850 Merck, Rahway, NJ, USA) under standard culture conditions (37 °C, 5% $CO_2$, and 95% humidity). To induce differentiation into macrophages, THP-1 cells were seeded in six-well plates and treated with phorbol 12-myristate 13-acetate (PMA, Cat. No. S7791 Selleck, Houston, TX, USA) at a final concentration of 100 nM for 48 h. Following differentiation, cells were washed with phosphate-buffered saline (PBS, Cat. No. G4202 Servicebio, Hubei, China) and allowed to rest in complete culture medium for 6 h prior to further treatment.

To establish an *in vitro* ALI model, THP-1-derived macrophages were stimulated with *Escherichia coli*-derived lipopolysaccharide (LPS, Cat. No. L2880 Sigma, St. Louis, MO, USA) at a final concentration of 1,000 ng/mL for 6 h, while the control group received an equal volume of PBS. Each experimental condition was conducted in triplicate to ensure reproducibility.

### CircRNA identification and RNase R treatment

Total RNA of cells was extracted from cells using TRIzol reagent (Cat. No. 15596018CN Thermo Fisher, Waltham, MA, USA) following the manufacturer's protocol, ensuring that samples were neither frozen nor chemically fixed prior to extraction. The quality and concentration of the extracted RNA was assessed using a NanoDrop spectrophotometer (Thermo Fisher, Waltham, MA, USA), measuring absorbance at 260 nm to determine RNA

concentration. The purity of RNA was evaluated by calculating the ratio of absorbance at 260 nm and 280 nm (A260/A280), with a ratio between 1.8 and 2.0 considered indicative of high-quality RNA.

For reverse transcription, one µg of total RNA was used to synthesize complementary DNA (cDNA) using a commercially available kit (Cat. No. RR036A Takara Bio, Shiga, Japan) on a ProFlex PCR System (Thermo Fisher, Waltham, MA, USA). We amplified cDNA using TB Green Premix Ex Taq™ II (Cat. No. RR42WR Takara Bio, Shiga, Japan) with specific primers.

For RNase R digestion, 1~10 µg of total RNA was incubated in a 20 µL reaction mixture containing 4 U RNase R per µg of RNA (NC1163864, Lucigen, Middleton, WI, USA) and two µL of enzyme buffer. The reaction was carried out at 37 °C for 15 min, followed by RNA purification using phenol-chloroform extraction to remove enzyme and reaction byproducts. The products were separated on a 2% agarose gel containing Nucleic Acid Red as a nucleic acid stain and visualized under UV illumination at 365 nm which were subjected to Sanger sequencing to identify the back splicing site.

## Quantitative real-time PCR

Quantitative real-time PCR (qRT-PCR) was performed to quantify the expression levels of circRNAs and mRNAs encoding proteins or cytokines using the QuantStudio 7 Flex Real-Time PCR System (Thermo Fisher, Waltham, MA, USA) using the SYBR Green (Roche Group, Basel, Switzerland) according to the manufacturer's protocol. Each reaction was prepared in a final volume of 20 µL, containing two µL of cDNA template, 10 µL of SYBR Green (Cat. No. 11200ES Yeasen Bio, Shanghai, China), and 0.4 µM of each forward and reverse primer. The cycling conditions included an initial denaturation at 95 ° C for 30 s, followed by 40 cycles of denaturation at 95 °C for 10 s and annealing/extension at 60 °C for 30 s. A melt curve analysis was performed to confirm the specificity of amplification. The primers for the candidate genes or circRNAs were listed in Table 1. *GAPDH* gene was selected as the internal reference gene. $2^{-\Delta\Delta CT}$ method was used to statistically analyze the RT-qPCR data. Each experiment included a minimum of three biological replicates per sample to ensure reproducibility.

## RNA oligo transfection

Cells were transfected with RNA oligo (synthesized by Tsingke, Shanghai, China) using Lipofectamine 2000 (Cat. No. 11668019 Thermo Fisher, Waltham, MA, USA), according to the manufacturer's instructions. Briefly, RNA oligos were mixed with Lipofectamine 2000 reagent in Opti-MEM (Cat. No. 31985070 Thermo Fisher, Waltham, MA, USA) and incubated for 20 min at room temperature to allow complex formation. The transfection mixture was then added to cells in a 6-well plate, and the medium was replaced with fresh complete medium 6 h post-transfection. Cells were harvested after another 48 h, unless differently specified. A detailed list of the RNA oligo used in this study is provided in Table 2.

**Table 1  Primers for candidate circRNAs/genes.**

| Name | | Sequence (5′–3′) |
|---|---|---|
| hsa_circ_0006354 | Forward | TCATCATCATCATGTGGTGGACA |
| | Reverse | AATTGAGAAGCGCCTGCCTG |
| hsa_circ_0035197 | Forward | ATTGTGCCTTTGGTCCTGGT |
| | Reverse | GTGGATACGATTATCTTGCTGTTG |
| hsa_circ_0001380 | Forward | GCAGCTTTGAAACAAGAAGAAGTT |
| | Reverse | GCCGTCGTCTTTTAGGAGCA |
| hsa_circ_0017092 | Forward | ATGATTCTGATTTTCAAGGGGTAG |
| | Reverse | TAATTCCAGCATACAAATCTCAGG |
| hsa_circ_0109021 | Forward | GCCACGCAAATCAAGACCTAC |
| | Reverse | CGCTGTGTCCAAGGTCGT |
| hsa_circ_0084615 | Forward | TTTATTAGAGACAAAGCATGGAGG |
| | Reverse | TACAGATGTCCAGACGCCCA |
| Human VAMP3 | Forward | ATGTGGGCAATCGGGATTAC |
| | Reverse | GCAGGTTCTAAGTCAAAAGTC |
| | Forward | GCGGCTTGAGAATGATGAATC |
| | Reverse | GTCGTATCGACTGCAGGGTCCG |
| hsa-miR-580-3p | | AGGTATTCGCAGTCGATACGACCCTAAT |
| | common-R | ACTGCAGGGTCCGAGGTATT |
| Human GADPH | Forward | AACGGGAAGCTTGTCATCAA |
| | Reverse | TGGACTCCACGACGTACTCA |
| Human IL-1β | Forward | ATGATGGCTTATTACAGTGGCAA |
| | Reverse | GTCGGAGATTCGTAGCTGGA |
| Human IFN-β | Forward | GCCATCAGTCACTTAAACAGC |
| | Reverse | GAAACTGAAGATCTCCTAGCCT |
| Human cGAS | Forward | TAACCCTGGCTTTGGAATCAAAA |
| | Reverse | TGGGTACAAGGTAAAATGGCTTT |
| Human STING | Forward | AGCATTACAACAACCTGCTACG |
| | Reverse | GTTGGGGTCAGCCATACTCAG |
| Human MDA5 | Forward | GAGCAACTTCTTTCAACCACAG |
| | Reverse | CACTTCCTTCTGCCAAACTTG |
| Human RIG-1 | Forward | CCAGCATTACTAGTCAGAAGGAA |
| | Reverse | CACAGTGCAATCTTGTCATCC |

**Table 2  Sequence of RNA oligo used.**

| Name | | Sequence (5′–3′) |
|---|---|---|
| si-circVAMP3 | Forward | CAUCAUCGGUGGUGGACAUUU |
| | Reverse | AAAUGUCCACCACCGAUGAUG |
| miR-580-3p-mimics | Forward | UUGAGAAUGAUGAAUCAUUAGG |
| | Reverse | UAAUGAUUCAUCAUUCUCAAUU |
| miR-580-3p-inhibitor | | CCUAAUGAUUCAUCAUUCUCAA |

## Lentivirus transfection

Lenti-circVAMP3 was synthesized and cloned by Genechem Co., Ltd. (Shanghai, China), and Lenti-NC was used as the negative control. For transfection, $2 \times 10^5$ cells per well were seeded into 24-well plates 24 h prior to transfection, ensuring that the cells reached was reached 70~80% confluence at the time of transfection. Transfection was performed using HiTransG A (Cat. No. REVG004 Genechem, Shanghai, China) according to the manufacturer's instructions. After transfection, cells were cultured for 48 h before being subjected to puromycin (Cat. No. 60209ES10, Yeasen, Shanghai, China) selection at a concentration optimized for the cell line. Cells were cultured under selection until no uninfected cells remained, and only the successfully transduced cells survived. The transduced cells were then harvested for subsequent experiments.

## Statistical analysis

All datasets were analyzed using GraphPad Prism (Version 10.3.0, GraphPad Software, LLC., Boston, MA, USA). Quantitative data, derived from a minimum of three independent biological replicates, were reported as mean $\pm$ standard deviation (SD) following validation of parametric assumptions. Briefly, normality was confirmed *via* Shapiro–Wilk testing, and variance homogeneity was assessed using Brown-Forsythe tests. Intergroup comparisons were conducted with two-tailed unpaired Student's $t$-tests, while multigroup analyses utilized one-way ANOVA preceded by Bartlett's test for variance homogeneity. Statistical significance was set at $^\star P < 0.05$.

# RESULTS

## circVAMP3 expression was significantly elevated in macrophage ALI model

We constructed an ALI model through LPS stimulation PMA-differentiated THP-1 macrophages. Subsequent qRT-PCR validation at the cellular level revealed distinct circRNA expression profiles, with concomitant elevation of *IL-1β* and *IFN-β* levels confirming successful model induction (Fig. 1A). Notably, *circVAMP3* (*circRNA.1_7777160_7778169*, *hsa_circ_0006354*) emerged as the exclusively upregulated circRNA in this macrophage-based ALI system (Fig. 1B). To further exclude the effect of VAMP3 linear RNA, we examined the linear RNA using sequence primers not present in circVAMP3 (in exon 5) and found that mRNA did not change significantly in the same model (Fig. 1C). Covalently closed circular structure was confirmed by qRT-PCR and gel electrophoresis on RNA either untreated (RNase R (-)) or treated (RNase R (+)) with the RNase R exonuclease. Linear RNA demonstrated significant alterations, whereas circRNA maintained structural stability (Figs. 1D, 1E). Then *circVAMP3* primer amplification product was purified and submitted to Sanger sequencing, and the results verified that back-splicing junction was elevated in amplified product (Fig. 1F), confirming the changed *VAMP3* expression was a circular RNA formed by reverse splicing of exons 3 and 4 rather than linear mRNA (Fig. 1G). Therefore, we speculated that circVAMP3 might be a potential regulator of pathophysiological processes in ALI/ARDS.

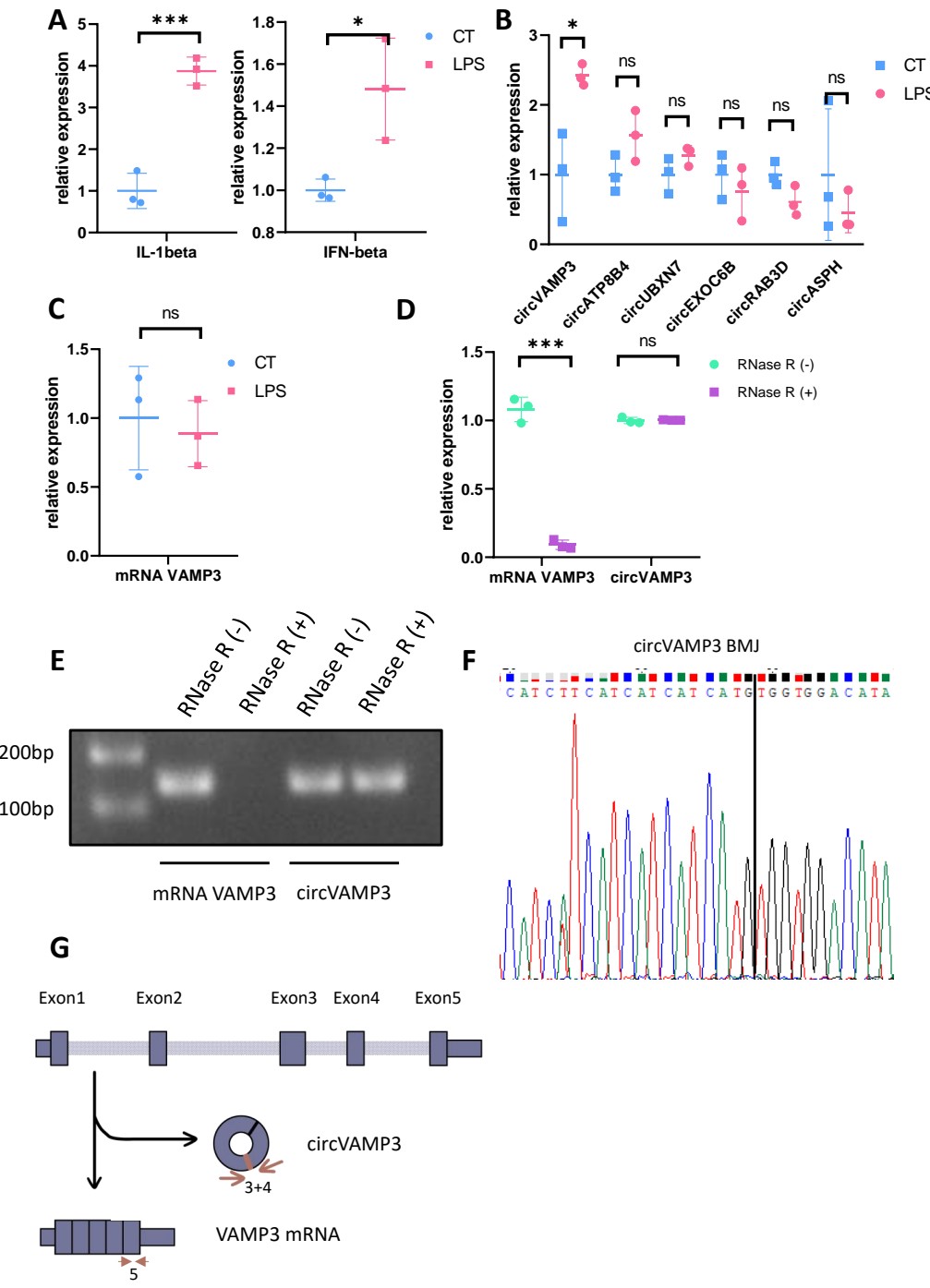

**Figure 1  Identification of circVAMP3.** (A) Relative expression of IL-1β and IFN-β in macrophage ALI model. (B) Relative expression of circRNAs in macrophage ALI model. (C) Relative expression of mRNA VAMP3 in macrophage ALI model. (D) Relative expression of VAMP3 RNAs after RNase R treated. (E) Gel electrophoresis of VAMP3 RNAs after RNase R treated. (F) Sequence of cyclization sites of circVAMP3. (G) The schematic diagram of circVAMP3. ***$P < 0.005$; *$P < 0.05$.

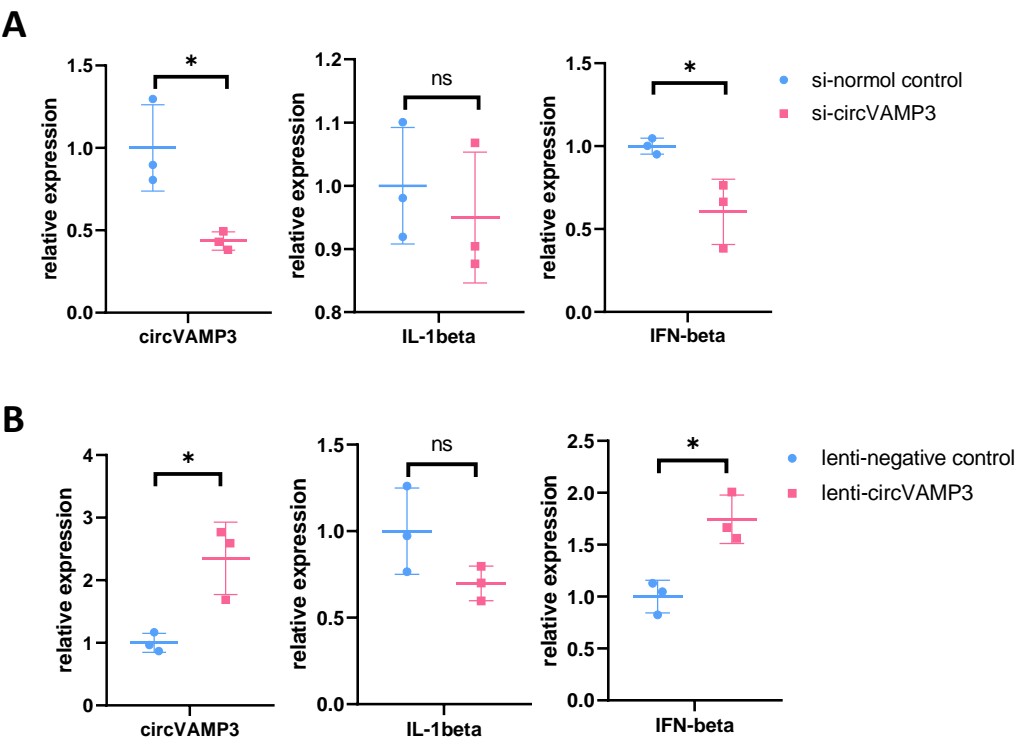

**Figure 2  CircVAMP3 regulates the expression of IFN-β.** (A) Relative expression of IL-1β and IFN-β in circVAMP3 knockdown model. (B) Relative expression of IL-1β and IFN-β in circVAMP3 overexpression model. *$P < 0.05$.

## CircVAMP3 regulates expression of IFN-β in macrophage ALI model

To elucidate the role of circVAMP3 in macrophage inflammatory responses during ALI pathogenesis, we knocked circVAMP3 down through siRNA and overexpressed through lentivirus. After successfully knockdown of circVAMP3, we found that IL-1β, a classical proinflammatory cytokine, remained unaffected, while IFN-β, a type 1 interferon primarily activated in macrophages, was significantly down-regulated under si-circVAMP3 (Fig. 2A). This specificity was further corroborated where lentiviral-driven circVAMP3 overexpression significantly potentiated IFN-β expression without altering IL-1β levels (Fig. 2B), suggesting that circVAMP3 may regulate the expression of IFN-β in macrophages in a certain specific way.

## *CircVAMP3* regulates the expression of IFN-β through STING in macrophage ALI model

To delineate how *CircVAMP3* regulates the expression of IFN-β, we studied the expression levels of major pathway proteins that regulate IFN expression on mRNA levels including cGAS-STING and MDA5/RIG-1 in macrophage ALI model. Our experimental results showed that in macrophage ALI model, the mRNA expression of cGAS and STING was significantly increased, and the expression of MDA5 was also increased, while there was no significant change in RIG-1 expression (Fig. 3A).

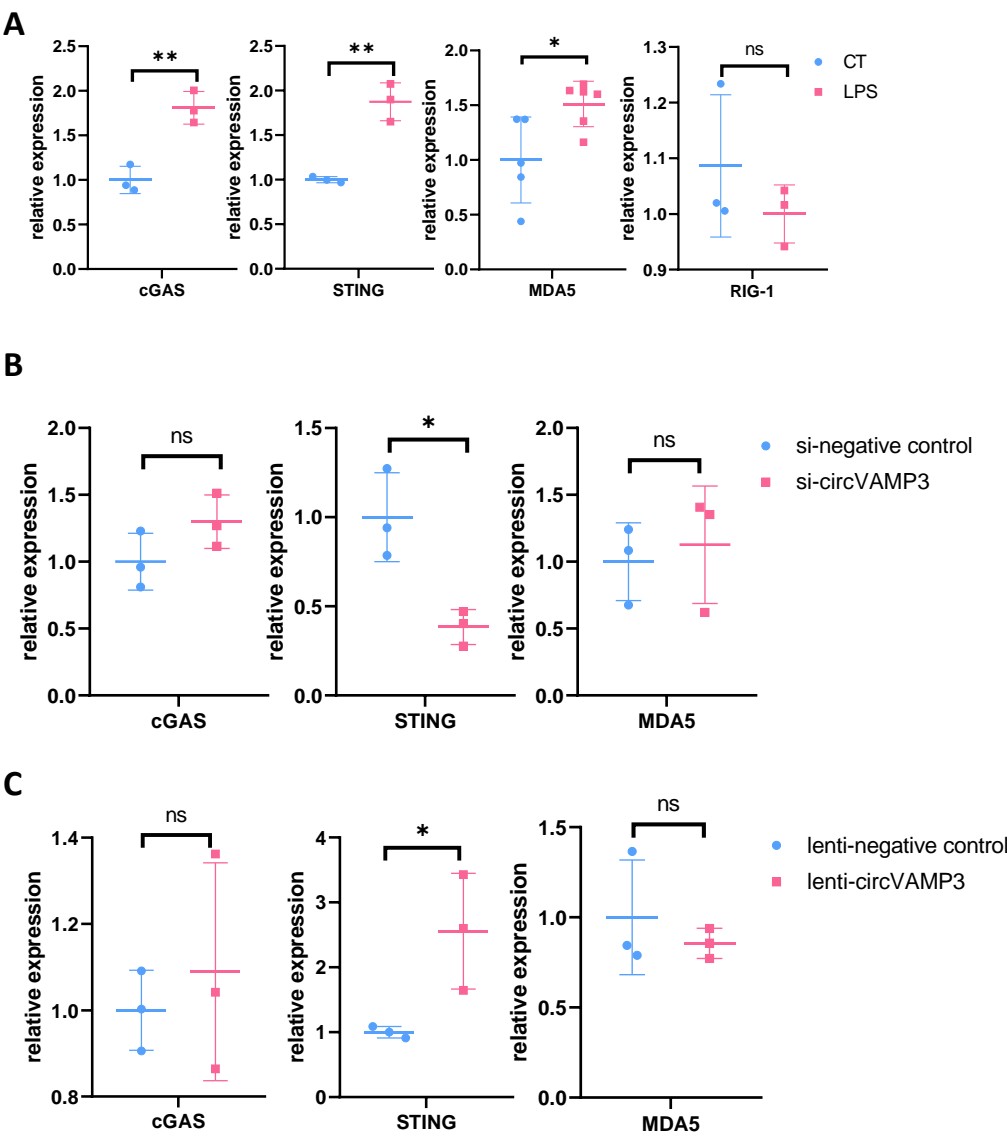

**Figure 3** **CircVAMP3 regulates the expression of IFN-β through STING.** (A) Relative expression of key protein regulating IFN-β in macrophage ALI model. (B) Relative expression of cGAS-STING and MDA5 in circVAMP3 knockdown model. (C) Relative expression of cGAS-STING and MDA5 in circVAMP3 overexpression model. $**P < 0.01$; $*P < 0.05$.

To further study which protein *CircVAMP3* targets, we examined the three proteins in cells which down-regulated and up-regulated *CircVAMP3*. We found that when transfected with si-*CircVAMP3* to down-regulated the expression level, only STING showed statistically significant downregulation synchronously, while cGAS and MDA5 did not change significantly (Fig. 3B). When we successfully transfected lenti-*CircVAMP3* to up-regulate *CircVAMP3*, only STING expression level increased with the up-regulation of *CircVAMP3*, and there was no statistical difference in cGAS and MDA5 expression

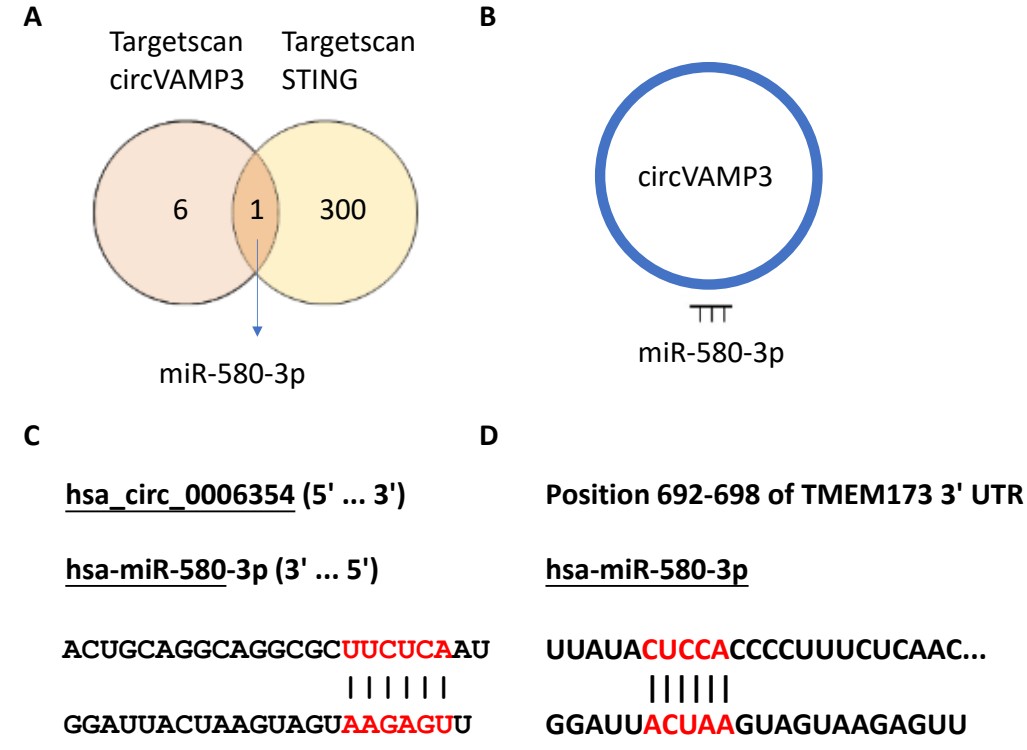

**Figure 4** **MiR-580-3p is the predicted key of circRNA.** (A) Venn diagram of miRNAs targeted by circVAMP3 and miRNAs targeting STING. (B) Schematic diagram of circVAMP3 sponge binding miR-580-3p. (C) Predicted sequence sites of circVAMP3 interaction with miR-580-3p. (D) Predicted sequence sites of miR-580-3p interaction with mRNA of STING.

level (Fig. 3C), which was consistent with the results of siRNA knockdown, suggesting *CircVAMP3* might selectively modulates STING transcription

### *CircVAMP3* regulates STING expression through miR-580-3p

In our previous experimental section, we had found that *CircVAMP3* might regulates IFN-β expression by influencing STING levels. Considering circRNA plays a role in the regulation of gene expression mainly through sponge binding specific miRNA, we speculated that there would be a miRNA regulated by *CircVAMP3* and regulating the mRNA of STING. We used the CircInteractome database (https://circinteractome.nia.nih.gov/) to predict target miRNA of *CircVAMP3* and TargetScanHuman (https://www.targetscan.org/vert_72/) to predict miRNA which can regulate mRNA of STING. We found a total of seven miRNAs that *CircVAMP3* could target and regulate, and 301 miRNAs that could target and regulate STING. Among the miRNAs obtained by databases, miR-580-3p is the only miRNA that is regulated by *CircVAMP3*, and regulate STING mRNA (Fig. 4).

To verify the role of miR-580-3p, we studied the expression levels in macrophage ALI models and *CircVAMP3* intervened models. The results showed that miR-580-3p was significantly down-regulated in macrophage ALI model compared with the control group and in the *CircVAMP3* intervention model, down-regulation of *CircVAMP3* expression

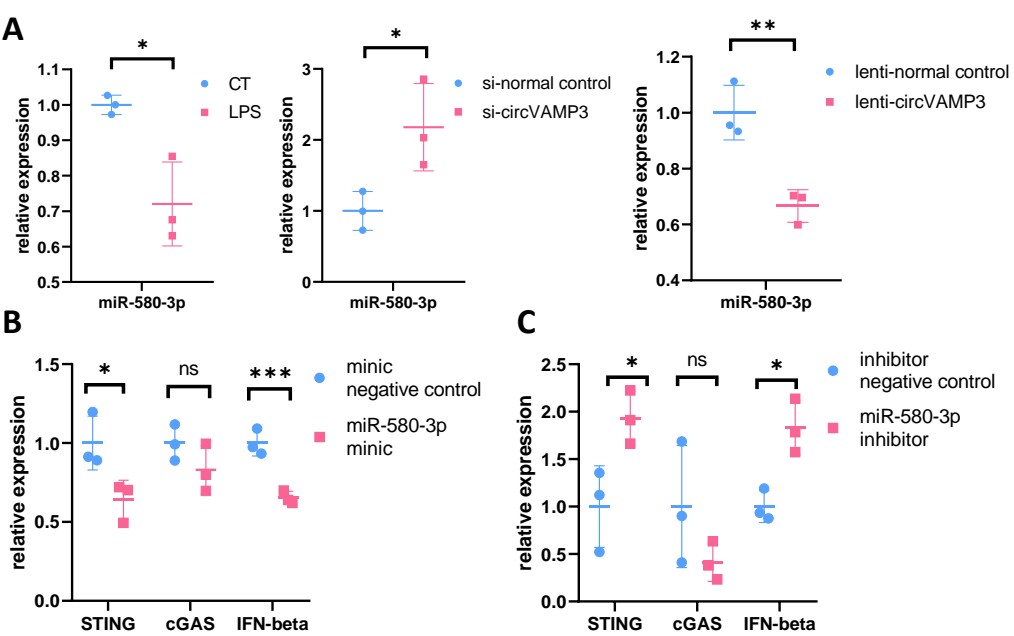

**Figure 5** **CircVAMP3 regulates the expression of STING/IFN-β through miR-580-3p. ***P < 0.005; **P < 0.01; *P < 0.05.** (A) Relative expression of miR-580-3p in ALI and circVAMP3 intervened models. (B) Relative expression of STING/IFN-β in miR-580-3p mimic model. (C) Relative expression of STING/IFN-β in miR-580-3p inhibitor model.

mediated by si-*CircVAMP3* led to the increase of miR-580-3p; up-regulation of *CircVAMP3* induced by lenti-*CircVAMP3* inhibited the expression level of miR-580-3p (Fig. 5A). The consistent results suggest that *CircVAMP3* can inhibit the expression level of miR-580-3p, and LPS can regulate miR-580-3p by regulating *CircVAMP3*.

After confirming the correlation between *CircVAMP3* and miR-580-3p, we further studied the correlation between miR-580-3p and STING. We regulated the effect of miR-580-3p through miRNA mimic and inhibitor to observe whether it had an effect on the expression of STING/IFN-β. Results showed that when miRNA concentration was increased by miR-580-3p mimic, STING expression was significantly down-regulated in mRNA levels, accompanied by simultaneous down-regulated levels of IFN-β. The expression level of cGAS was not affected (Fig. 5B). However, after the biological effects of miR-580-3p were inhibited by inhibitor, the expression level of STING was increased, accompanied by the simultaneous increase in the expression level of IFN-β (Fig. 5C), suggesting that miR-580-3p could bind STING mRNA. So far, our experiments have proved that *CircVAMP3* can inhibit the miRNA effect of miR-580-3p in macrophage ALI model, thereby up-regulating the expression level of STING and promoting the expression of IFN-β.

## DISCUSSION

Circular RNA is a circular single-stranded RNA structure formed through alternative back-splicing mechanisms during post-transcriptional processing. The complex and varied splicing mode leads to the formation of circRNA with different sequences by splicing RNA structures encoding the same genes, which act as sponges to bind different miRNAs and thus regulate life activities. The *CircVAMP3* in our research is a circular structure obtained by reverse splicing of one of the sequences encoding the VAMP3 gene. In previous experimental studies, *CircVAMP3* was found to be highly expressed in lung rhabdomyosarcoma cells, playing an important role in promoting tumor cell proliferation (*Rossi et al., 2021*). In addition, we also found increased expression of *CircVAMP3* in LPS-stimulated macrophage lung injury model, suggesting that *CircVAMP3* may also play an important role in acute inflammation and immunity. As a key molecular pathway involved in inflammatory apoptosis and innate cellular immunity, cGAS-STING has been shown to be strongly correlated with LPS-induced lung injury models (*Ning et al., 2020*). Our study verified the relationship of *CircVAMP3*/miR-580-3p/STING, providing a new target and direction for regulating STING/IFN-β interferon pathway.

CircRNAs is a treasure with great excavation value. From sequence lncRNAs (*Nojima & Proudfoot, 2022*) to circRNAs (*Chen, 2020*) and miRNAs (*Gebert & MacRae, 2019*) formed by variable shear, ncRNAs weave a complex and ordered network in the process of cell life activities, precisely regulating changes in life activities in different aspects. Advances in information technology and algorithm technology provide solid technical support for the prediction of ncRNAs interaction and accelerate the research of ncRNAs, but at the same time, a large number of ncRNA studies are superficial and cannot become valuable clinical diagnosis and treatment targets. In earlier studies, some circRNAs that are highly expressed in ARDS patients were reported, with the corresponding miRNA targets and further downstream gene targets searched by software algorithms and a complex circRNA-miRNA-mRNA interaction network obtained. However, these studies lack follow-up and further experimental studies *in vitro* or *in vivo* (*Guo et al., 2021*; *Wang et al., 2021*). The diversity of ncRNAs targets and action sites, as well as the spatiotemporal specificity of expression or regulation, make the results obtained by algorithms unable to be directly applied to the clinic. This process requires more convincing direct evidence. In recent years, most of the experimental studies on human circRNA focus on the direction of tumor, and few people pay attention to the inflammation of the lung such as ARDS or the diseases with certain immune correlation. Based on computer analysis, a recent study verified the mechanism of action of Circ_000149 through cell models and animal models of lung injury caused by sepsis, providing a new idea for the targeted regulation of SOX6 (*Hu et al., 2023*). In contrast, our experiments were mainly conducted in the lung injury model of macrophages, which verified the existence of a circRNA-miRNA-mRNA pathway in macrophages, and proved that *CircVAMP3*/mi-580-3p/SITNG/IFN-β plays an important role in immune cells.

As an important part of the innate immune system, cGAS-STING plays a crucial role in the secretion and regulation of IFN-β induced by double-stranded DNA. IFN-β has a certain

basic expression level in most human cells, mainly through increasing the level of major histocompatibility complex (MHC) on the cell surface to regulate and mediate the cleavage of impaired cells. At the same time, the secretion of chemokines and cytokines by cells is increased to further recruit immune cells and inflammatory cells to the local inflammatory response and coordinate the immune response (*Ng et al., 2016*; *Wang et al., 2020*). In acute lung injury models, cGAS-STING has been reported to regulate the expression of type 1 interferon through TBK1 (*Wang et al., 2020*), IL-1β and other inflammatory factors which play an important role in lung injury. Experiments have also verified the existence of cGAS-STING pathway in the macrophage model (*Vincent et al., 2017*). The changes of STING and IFN-β in the same direction in our experiment are consistent with the results reported in previous studies. IL-1β levels did not change with STING protein levels, suggesting that other inflammatory pathways may affect the regulation of NLRP3 or IL-1β in macrophage lung injury models.

While our study provides novel insights into the *CircVAMP3*/miR-580-3p/STING axis in ALI pathogenesis, several limitations should be noted. First, although bioinformatics predictions and functional rescue experiments support *CircVAMP3*'s role as a miR-580-3p sponge, direct binding validation (*e.g.*, RNA pull-down or dual-luciferase reporter assays) remains to be performed. Second, our findings are derived mostly from *in vitro* macrophage models, future studies should validate this regulatory axis in animal ALI models and clinical ARDS cohorts. Third, the functional diversity of circRNAs, including interactions with RNA-binding proteins or modulation of parental gene expression, was not explored, leaving open the possibility of alternative mechanisms. Finally, while we focused on IFN-β regulation, the broader impact of *CircVAMP3* on macrophage polarization and crosstalk with other immune cells warrants investigation. Addressing these limitations will strengthen the translational relevance of targeting *CircVAMP3* in cytokine storm-related diseases.

## CONCLUSIONS

Our experiment demonstrated the effect of *CircVAMP3* on miR-580-3p, which leads to upregulated STING protein expression and subsequent activation of IFN-β production, establishing a novel regulatory axis in the STING/IFN-β signaling pathway. Importantly, we identified miR-580-3p as a previously unrecognized mediator of inflammatory factor regulation, with our findings suggesting its critical role in modulating inflammatory storm dynamics. These mechanistic insights also propose target of the *CircVAMP3*/miR-580-3p/STING axis for STING/IFN-β pathway modulation and, providing a foundation for precision therapeutic development in cytokine storm-related diseases.

### Funding

This work was supported by the National Natural Science Foundation of China (No. 81870062) and the National Nature Youth Foundation (No. 81900038). The funders had

no role in study design, data collection and analysis, decision to publish, or preparation of the manuscript.

## Grant Disclosures

The following grant information was disclosed by the authors:
The National Natural Science Foundation of China: No. 81870062.
The National Nature Youth Foundation: No. 81900038.

## Competing Interests

The authors declare there are no competing interests.

## Author Contributions

- Zheng Zhou performed the experiments, analyzed the data, prepared figures and/or tables, and approved the final draft.
- Shuyang Chen analyzed the data, authored or reviewed drafts of the article, and approved the final draft.
- Yajun Wang analyzed the data, prepared figures and/or tables, and approved the final draft.
- Shujing Chen analyzed the data, authored or reviewed drafts of the article, and approved the final draft.
- Jinjun Jiang conceived and designed the experiments, authored or reviewed drafts of the article, and approved the final draft.

## DNA Deposition

The following information was supplied regarding the deposition of DNA sequences:

The GADPH is available at GenBank: Chromosome 12, NC_000012.12 (6534517..6538371).

The VAMP3 is available at GenBank: Chromosome 1, NC_000001.11 (7771296..7781432).

The IL-1β is available at GenBank: Chromosome 2, NC_000002.12 (112829751..112836779, complement).

The IFN-β is available at GenBank: Chromosome 9, NC_000009.12 (21077104..21077942, complement).

The cGAS is available at GenBank: Chromosome 6, NC_000006.12 (73423711..73452297, complement).

The STING is available at GenBank: Chromosome 5, NC_000005.10 (139475533..139482758, complement).

The MDA5 is available at GenBank: Chromosome 2, NC_000002.12 (162267074..162318684, complement).

The RIG-1 is available at GenBank: Chromosome 9, NC_000009.12 (32455302..32526196, complement).

The hsa-miR-580-3p is available at GenBank: Chromosome 5, NC_000005.10 (36147892..36147988, complement).

The hsa_circ_0084615 is available at circBase: chr8:62593526-62596747.

https://www.circbase.org/cgi-bin/singlerecord.cgi?id=hsa_circ_0084615

The hsa_circ_0109021 is available at circBase: chr13 20279801 20279971.

https://www.circbase.org/cgi-bin/singlerecord.cgi?id=hsa_circ_0109021

The hsa_circ_0017092 is available at circBase: chr1:235963619-235964397.

https://www.circbase.org/cgi-bin/singlerecord.cgi?id=hsa_circ_0017092

The hsa_circ_0001380 is available at circBase: chr3:196118683-196129890.

https://www.circbase.org/cgi-bin/singlerecord.cgi?id=hsa_circ_0001380

The hsa_circ_0035197 is available at circBase: chr15:50330964-50339661.

https://www.circbase.org/cgi-bin/singlerecord.cgi?id=hsa_circ_0035197

The hsa_circ_0006354 is available at circBase: chr1:7837219-7838229.

https://www.circbase.org/cgi-bin/singlerecord.cgi?id=hsa_circ_0006354

## Data Availability

The raw measurements are available in the Supplementary File.

## Supplemental Information

Supplemental information for this article can be found online at http://dx.doi.org/10.7717/peerj.19573#supplemental-information.

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
