# Peer review of "CircVAMP3 promotes acute lung injury progression by regulating the miR-580-3p/STING/IFN-β axis in macrophages"

_PeerJ, doi:10.7717/peerj.19573_

## Round 0.1 · original submission · Major Revisions

The authors are advised to revise the manuscript as per reviewers comments.

**Language Note:** The review process has identified that the English language must be improved. PeerJ can provide language editing services - please contact us at [email protected] for pricing (be sure to provide your manuscript number and title). Alternatively, you should make your own arrangements to improve the language quality and provide details in your response letter. – PeerJ Staff

Reviewer 1 ·

Basic reporting

figures are not professional. more details should be included, For instance, for the mimic NC, and inhibitor NC, inhibitor etc. the names should be written clearly. And for rt-qPCR, the replicates should be shown as dots and in the figure legend, how many replicates are used should be explained clearly, like n=3 or n=4.

Experimental design

there is a critical drawback of the experimental design: For siRNA knockdown in THP-1 derived macrophages, transfection of siRNA into THP-1 cells will inevitably activate the TLR3 and RIG-I pathways as siRNA will be recognised as a exogenous material. Therefore, the data in Figure 2B is less controlled. A lentiviral transduced knockdown should be used instead of siRNA transfection.

Validity of the findings

1, for the statement, at line 124 : we examined the linear RNA using sequence primers not present in circVAMP3. This should be shown as a scheme in Figure 1 to show the primer targeting sites. Also, the primer sequence should be provided in method section.

2, if circVAMP3 served as a sponge to bind the miR-580-3p, there is a lacking of the direct binding data between circVAMp3 and miR-580-3p.

Additional comments

the conclusions that circVAMP3 regulates miR-580-3p/STING/IFN-b axis is a over-interpretation of the data. The causality between those RNA and proteins is ambiguous.

·

Basic reporting

The manuscript is poorly written and requires significant language editing to improve clarity and readability. Issues include incorrect verb tenses, awkward phrasing, and grammatical errors.

The structure conforms to typical scientific standards, but sections like the introduction and discussion lack flow and coherence. The introduction could have been more concise, focusing on the knowledge gap and study rationale.

The manuscript references relevant literature, but some claims lack sufficient citations, especially regarding the novelty and significance of circVAMP3 in ARDS.

Figures need clearer labeling and higher resolution.

There are no hypotheses provided in the manuscript or the lacunae.

Experimental design

The methods are generally well-described, but some areas lack detail. For example, the RNA transfection section should specify the exact sequences used, and the statistical analysis section should include information on how normality was assessed and if corrections for multiple comparisons were applied. Additionally, verification of circular RNA should include treatment with RNase R to confirm the circular nature of circVAMP3. The absence of RNase R treatment in this study is a significant omission that should be addressed to validate the presence of circular RNA in the samples.

The study includes appropriate controls, but the number of biological replicates should be clearly stated. Details on randomization and blinding, if any, should be included to strengthen the rigor.

Validity of the findings

The data appear robust, with consistent findings across different experimental approaches. However, more detailed statistical analyses would enhance confidence in the results.

The conclusions are generally supported by the data but should be more cautiously framed. The manuscript suggests circVAMP3 as a potential therapeutic target, which requires validation in in vivo models before such claims can be substantiated.

Additional comments

The present study explores a novel regulatory mechanism in ALI, providing new insights into circRNA functions. The experimental design is somewhat comprehensive, covering multiple molecular techniques.

However, language issues hinder comprehension. The manuscript lacks a clear statement of limitations and could benefit from additional in vivo validation to support the therapeutic potential of circVAMP3. The lack of RNase R treatment to confirm the circular nature of circVAMP3 is a notable methodological gap.

---

## Round 0.2 · Minor Revisions

Thank you for submitting your manuscript to PeerJ. The peer-review process has been completed, and we kindly ask the authors to address the minor comments provided by Reviewer 2.

Reviewer 1 ·

Basic reporting

authors have addressed my concerns

Experimental design

authors have addressed my concerns

Validity of the findings

authors have addressed my concerns

Additional comments

authors have addressed my concerns

·

Basic reporting

I have seen the changes incorporated by the authors. They did well to incorporate the suggested changes and performed additional experiments to answer the reviewers questions.

Experimental design

The experimental approach was well designed, and after suggested modifications, the data looks more robust.

Validity of the findings

Though the authors have incorporated necessary changes, they failed to provide the gel images of circular rna circVAMP3 after RNaseR treatment, which is necessary to confirm the circular nature of circVAMP3, in addition the origin of samples used for sanger sequencing is also missing.

---

## Round 0.3 · accepted · Accept

The comments raised by the reviewers have been addressed. My recommendation is to accept.

·

Basic reporting

Everything looks good now.

Experimental design

Well designed and conducted.

Validity of the findings

The findings of the study is valid and relevant.